# Communicating Unexpected and Violent Death: The Experiences of Police Officers and Health Care Professionals

**DOI:** 10.3390/ijerph191711030

**Published:** 2022-09-03

**Authors:** Diego De Leo, Benedetta Congregalli, Annalisa Guarino, Josephine Zammarrelli, Anna Valle, Stefano Paoloni, Sabrina Cipolletta

**Affiliations:** 1Australian Institute for Suicide Research and Prevention, Griffith University, Brisbane, QLD 4122, Australia; 2Slovene Center for Suicide Research, Primorska University, 6000 Koper, Slovenia; 3De Leo Fund, 35137 Padua, Italy; 4Autonomous Syndicate of Police (SAP), 00184 Rome, Italy; 5Department of General Psychology, University of Padua, 35100 Padua, Italy

**Keywords:** death notification, external cause, police officers, doctors, nurses, training, forms of support, impact on recipients

## Abstract

Background: The notification of unexpected and violent death represents a challenging experience for police officers and health workers. These professionals are exposed to very intense emotions during this task. Aim: We aimed to investigate the degree of preparation, and the emotions and attitudes of police officers and health professionals while communicating such a death. Method: An ad hoc online questionnaire was created and disseminated through Qualtrics software. The participants were recruited through the institutional channels of Police, the College of Physicians, the ONG De Leo Fund and the Department of General Psychology of the University of Padua. In this qualitative study, thematic content analysis was used to examine the responses. Results: A total of 155 individuals participated in the study (44 females, 111 males): 102 individuals were police officers, 23 were doctors and were 30 nurses. Five main themes were identified: (1) how the communication took place; (2) the experiences during the communication; (3) the difficulties encountered; (4) coping strategies, and (5) forms of support. Most communications were performed in person, and most represented an intense emotional experience for the notifiers. There is a generalised lack of specific preparation and training for this practice. The recipient’s characteristics (culture of origin, language, age, health conditions, psychological vulnerability) may add to the difficulties of the notification process. Professionals unload their tension by practicing sport, relying on their hobbies or interacting socially. The presence of other colleagues during and after the death notification is usually able to alleviate the burden of the communication. Conclusions: Communication modalities can have a profound impact on the recipients and intensify the trauma of the loss; however, they also have the possibility of mitigating it. The notification of a violent and unexpected death remains a difficult and challenging task for the notifier, which is potentially stressful and emotionally charged. The topic is of great relevance and more research should be promoted in this area.

## 1. Introduction

In this article, by ‘traumatic loss’ we mean all those circumstances in which the death of a loved one occurs suddenly and violently, such as in cases of suicide, murder, road accident, accident at work and natural disaster [1]. These are unpredictable and sudden deaths as the individual who mourns them generally did not have the opportunity to mentally anticipate them [2].

Traumatic deaths usually have significant consequences on the mental health of those who remain [2], which can complicate the grieving process [3]. Various factors can hinder or favour the processing of the loss [4]. Additionally, the quality of the communication of death must be considered, as the way in which the notification is made can affect the process of adaptation to the loss [5].

Notification is the official communication of the death of a loved one, conducted mostly by a doctor, nurse or police officer [6] or, in some cases, by religious figures [7]. Professionals in charge of communication should be fully aware of the importance of having an attitude appropriate to the delicate circumstance [8], since sensitive communication can contain the person’s reactions [9]. For this reason, it is essential to work on the communication skills of the professionals involved so as to be able to reduce, at least in part, both the pain of the relatives and the stress that the execution of this task entails [10].

The communication of an unexpected and violent death forces the people involved to live very intense moments: for the recipients (family and friends), the moment of notification represents a watershed between a before (the life that was) and an after (the life that will be without the loved one) [6]. This is a circumstance that often leaves an indelible mark over time, of which a particularly clear memory is preserved [11]. In addition to personal discomfort, the operators in charge of giving the tragic news must also learn to manage the emotional disturbance manifested by the recipients. Indeed, communicating an unexpected and violent death can provoke reactions of shock and anguish in the recipient [12], prolonged silences or strong emotional outbursts, accompanied by screams and manifestations of anger [13], dissociation and depersonalization [14].

It often happens that professional figures have to interact with family members in contexts such as hospital wards, the site of the accident or the home of the notified persons. In such circumstances, the assessment of needs, attitudes and beliefs and the building of the relationship with the person tend to occur simultaneously rather than sequentially [15]. The lack of previous knowledge, the often inadequate spaces [16], the limited time available and the cultural and age differences between interlocutors [17] make the notification process very challenging. In addition to managing these aspects, the notification of traumatic death entails a considerable emotional and stress load for the professional figure involved [18], which is also related to the full awareness that the notification will forever alter the lives of who receives it [17]. Practitioners may feel unprepared [7], feel discomfort [19], guilt or fear of being blamed [20], anxiety and distress [13], sadness and sorrow for the notified person’s pain, and frustration [7,18]. In addition, somatic reactions such as tachycardia or heart rate instability may occur [3], but also insomnia and fatigue [21]. Understandably, repetitive exposure to this task can lead to a greater likelihood of emotional ‘exhaustion’ [22,23]. These are just some of the reasons why it is important for professionals to acquire adequate skills and strategies, in order to improve their sense of self-efficacy [24].

This article presents the initial results from the perspective of the notifiers through a qualitative interpretative research design [25], which was considered the most suitable method for capturing the experience of the individuals and their particular perspectives on the situation they were involved in [26]. In particular, we aimed to investigate the degree of preparation of the professionals most often involved in this difficult task to communicate an unexpected and violent death. We gave space to the narratives of doctors, nurses and police officers. The experience of the communication and subsequent moments, the strategies adopted, the training and support received to deal with communication were the main areas investigated.

## 2. Methods

The participants were recruited through the institutional channels of the Autonomous Police Syndicate (SAP) and the College of Physicians of the provinces of Rovigo and Belluno, which advertised the investigation, thus promoting the participation of police officers and health professionals. Furthermore, the social channels of ONG De Leo Fund and the Department of General Psychology of the University of Padua were used to widen the recruitment. To collect the personal expression of the experiences lived by professionals that have communicated an unexpected and violent death during their duties, an ad hoc online questionnaire was created and disseminated through the Qualtrics software. The questions were created ad hoc, because it allowed us to more freely investigate areas of interest. In addition, we chose to administer the questionnaire online in order to respect the criterion of economy and reach a sample as large as possible [27]. The questionnaire was also designed to minimise the danger of re-traumatisation of the participants, who were given the option of abandoning the questions if they realised the questionnaire was too intrusive and could close the online page without further explanation. Useful contacts were provided to the participants at the end of the questionnaire in case the questions had triggered intense and upsetting emotions.

The questionnaire consisted of 11 items collecting demographic information and 10 open questions to investigate the experiences of the professionals. For example, they were asked, *“What did you experience while communicating?”*, “*What were the main difficulties you encountered?*” and “*What measures did you take to find relief from the burden of communication*?” (see Table 1). Participation in the study was on a voluntary basis. All the subjects involved in the study were asked to provide informed consent.

A total of 155 Italian individuals participated in the study (44 females, 111 males, mean age = 47.74; SD = 9.46). Among the participants, there were 102 police officers (11 females; 91 males; mean age = 48.52; SD = 7.25); 23 doctors (10 females; 13 males, mean age = 52.20; SD = 11.43) and 30 nurses (23 females; 7 males; mean age = 41.27; SD = 11.86). Table 2 shows the demographic information relating to the sample.

The data was collected from January 2021 to July 2021, and according to the standard practice for qualitative research, the data collection and analysis of the questionnaires proceed simultaneously.

The response obtained in the questionnaires were analysed through thematic content analysis (TCA). Initially, the thematic analysis was carried out following the six-phase method proposed by Braun and Clarke [28]; it was performed by two authors and supervised by a professor (SC), who is also a trained coder. The researchers initially read and re-read the data set until the depth and breadth of the content became familiar. Then, the initial codes were generated from the data. Subsequently similar codes were grouped into similar sub-themes, which in turn, were grouped into general themes, and the relationships between the observed themes were identified. The results were revised to evaluate if all the themes had sufficient supporting data with appropriate extracts and to judge if they met the internal homogeneity and external heterogeneity criteria. Finally, the themes were refined and named to indicate what topics they captured. This process led to the creation of a codebook, which was applied to all the responses. A consensus on themes was achieved by continuous discussion between the team of researchers. At the end of the TCA we quantified the data by frequency [29]. The coding of the textual material was carried out manually with the aid of colored markers, using a “paper and pencil” procedure.

## 3. Results

The TCA [29] of the narratives provided by the participants led to the identification of five themes: (1) how the communication took place; (2) the experiences during the communication; (3) the difficulties encountered; (4) coping strategies, and (5) forms of support (Table 3).

### 3.1. How Did the Communication Take Place

Among the 155 participants, 117 reported they had given the death notification in person: they underlined the importance of being able to have direct contact with the notified person: *“The three times I found myself communicating death, I always did it in person. As difficult as it is, I think and believe that it is the only way to do it and the most appropriate too.”* (P15, M, police officer).

Nine participants reported the death over the phone. The use of this means was justified by causes of *force majeure*, such as the geographical distance of the relatives: *“It has to be done only in very few cases (geographical distance of the relative to whom the communication must be provided) and when one is obliged by causes of force majeure, otherwise it is more appropriate to do it in other ways.”* (P31, M, police officer).

Another 21 participants had given the death report both over the phone and in person. In those cases, the professional first contacted family members by telephone, then summoned them to the hospital and only on their arrival proceeded with the formal communication: *“When I worked in ED family members were warned by phone of a serious and traumatic event and were asked to reach our facility. Never directly the news on the phone.*”(P115, M, nurse). The remaining eight participants did not clarify the type of contact they used to give the death notification.

The method used to notify the traumatic death influences the experience of professionals; a nurse reported: *“Not being present when I gave the tragic news helped me not to get caught up in emotion”* (P149, M, nurse) and, another nurse: *“It is easier for the operator to communicate the news on the phone because you do not see the face of the interlocutor, but this modality makes it difficult to guide the different phases faced by the bereaved person.”* (P131, F, nurse).

In terms of approaches and styles (what to say and how), opinions and behaviours diverged significantly: 26 participants said that it was important to explain in detail the sequence of events that led to the death of the loved one. In contrast, 13 professionals considered it as more appropriate to avoid the details because they were not explicitly requested by family members. Nine participants opted for dry and fast communication, whereas 22 professional figures preferred to communicate the death gradually:

“*It is useful to communicate the death in stages, for example by taking short breaks and not giving the news immediately. It is useful for relatives to understand what has happened to their loved one and for them to conclude the sentence, so as to be able to offer them at least some support.*”(P147, M, doctor).

Finally, 25 participants chose to communicate in an ‘extemporaneous’ way, that is, without implementing any specific measure and sometimes even letting the notified know the news in advance. This choice is part of what can be defined as “non-communication communication”, or those cases in which the notification has not been fully expressed but the tragic reality has been left to be deduced by the recipient.

### 3.2. The Experiences during the Communication

This “lived” theme consists of two sub-themes: emotions and reactions. As for emotions, 68 professional figures said they felt a sense of inadequacy with respect to the assigned task, not feeling up to the task. Twenty-four participants said they felt a sense of injustice, as the death occurred not only unexpectedly but also prematurely, and was violent. Twelve professionals spoke about the fear of possible long-term effects on themselves, as a result of the continuous exposure to the theme of death.

In regards to their reactions, ten participants could not control their crying after the notification, 15 experienced somatic reactions such as chills down the back, intense physical exertion, an adrenaline rush, a lump in the throat, emptiness, palpitations, intense sweating and nausea. Seven participants reported experiencing shock, depersonalization, stress and estrangement.

“*I don’t think I know how to report the drama of those moments; I know I am a professional and as such I have a role, but being a person and a mother and seeing the agony of parents who have just lost their children, especially through violent and therefore dramatically sudden death, cannot be described. [..] I lived for a week in a state that I could define as a “trance” even thinking about the reaction of that mother*.”(P121, F, doctor).

### 3.3. The Difficulties Encountered

Among the difficulties that participants reported were those related to the personal and family characteristics of the notified person: for 13 professionals the advanced age of the notified person and the fear of a possible illness, the degree of kinship between the deceased person and family member and the state health of the recipients are all aspects that made the notification of death difficult: *“It was difficult to communicate the news to parents with psychiatric problems, a history of alcohol abuse, and social problems.”* (P121, F, doctor). In addition, 58 participants said they had difficulties related to adapting to their interlocutor and their culture of origin. The difficulties concerned mainly linguistic factors, and highlighted how the culture to which one belongs can also play an important role in the ways in which pain is expressed: “*Every situation, every family and every circumstance of death is unique; for this reason you have to be able to adapt*.” (P101, M, police officer).

Among the emotional difficulties, 33 professionals said that the very intense reactions to the notification made the assignment quite complex: *“I will never forget when the mother collapsed to the ground. In those circumstances I felt really useless.”* (P129, F, nurse). Many participants (*n* = 75) confessed finding the management of their emotions difficult when facing the pain of family members who receive the news. They reported that the search for the right balance between involvement and detachment was particularly problematic. In this sub-sample of professionals, two opposing ways of thinking were observed: 15 participants believed that trying to identify with their interlocutor, however difficult, is a resource that allows them to understand the needs of the person and, therefore, to conduct the notification of death in a respectful manner. However, 13 other professional figures communicated that maintaining a certain detachment can guarantee greater clarity in carrying out the task: *“To facilitate the reception of the news it was useful to put myself in the third person, using detachment.*” (P33, F, police officer).

### 3.4. Coping Strategies

Compared to the fourth area identified, 32 participants stated that once the notification task was complete, they needed moments of distraction by playing sports, dedicating time to their hobbies or just resting long hours: *“A relief valve that I often use to dispose of this type of weight is to play sports: it allows me to focus on my body. Another thing that I find useful in these circumstances is chatting with relatives and friends. Distracting myself from what has happened helps me a lot.”* (P14, M, police officer).

Sixty-five participants reported having implemented avoidance strategies. Specifically, 49 participants stated that they did not use any specific measures to stop feeling the burden of the communication. For example, *“We have difficulties in speaking freely and openly, because we are afraid of suspension,”* (P13, M, police officer) or *“I have not adopted any precautions to take away the weight of the communication. I kept it all inside. I’ve never looked for real support. […] The feeling that this type of position leaves is very intense; it is very difficult for me to open up.”* (P92, M, police officer).

### 3.5. Forms of Support

As described by the professional figures, the following categories were identifiable: training, support sought and support received. Of the 155 survey participants, 107 said they had not received any training in how to conduct a death notification: 

“*In all my professional training and clinical practice no one has helped me to have preparation or techniques to deal with this type of conversation; no company cares about how healthcare professionals react emotionally to giving this type of news. It is considered as part of the job; it is not talked about, and it is taken for granted that the healthcare professional is not affected by anything that concerns the death of a patient*.”(P144, M, doctor).

However, not many individuals (*n* = 7) among the professional figures who did not receive any training felt it was needed: *”It is important that professional figures can rely on homogeneous guidelines, and not just on the common sense of the individual.”* (P117, M, male nurse). In addition, four participants said they would not suggest any training. A State Police officer said: 

“*Unfortunately, in my view, there are no teachings suitable for reporting such heartbreaking news. Especially when you have a couple of parents in front of you—and you know what it does mean to have children—I believe that in these situations formal knowledge is not necessary; what is essential is to maintain a substantially human attitude*.”(P69, M, police officer).

Other participants (*n* = 39) reported that, despite not having received specific training, in facing this difficult assignment they relied on previous experiences, such as having already given a death notification, having observed more experienced colleagues giving this type of news or having thought back to when they had been the notified person. One doctor, for example, said: *“No, I have not received any specific training. I think I have referred to my personal experiences as a recipient.*” (P119, M, doctor).

With regard to support sought, 17 professional figures reported that they found the informal support received by colleagues, friends and relatives very useful to help normalize what they felt at an emotional level: *“More experienced colleagues have been of great help to me; the closeness of my family and friends was also very important. This profession is very beautiful but it is also very stressful. Well, with them I felt free to cry…”* (P145, F, nurse). Many more participants (*n* = 66) said that it was important to be supported by another colleague or other professional (doctor, nurse, psychologist, religious figure) as they were helpful during the communication for support and before the notification for an exchange of views. A police officer wrote: *“It was helpful for me to be with others or with another colleague. This allowed me to discuss both before going to family members and once the assignment was over. Being accompanied by a colleague made me feel less alone in that difficult moment and allowed me to feel more courageous.”* (P14, M, police officer).

## 4. Discussion

This qualitative study provides an overview of the experience primarily of Italian police officers and also health care workers who in their careers have communicated violent and unexpected death. This qualitative investigation reveals, in addition to emotional aspects, the training received, support, and coping strategies implemented. The participants are mostly male police officers and health care workers are mostly female, consistent with the gender distribution of police officers and health care workers in the Italian population [30,31]. Most of the notifications were made in person, while the telephone was a medium that sometimes made the task easier and sometimes made it even more difficult. The professional figures adopted different communication styles to deliver the news. The task represented an intense emotional experience on the part of the notifiers, who felt a sense of injustice, helplessness, inadequacy and concern in performing the task. The characteristics of the recipient (culture of origin, language, age, health condition, psychological vulnerability) and the management of the notifier’s emotional aspects may have increased the difficulties of the notification process. Some professionals discharged their tension by playing sports, pursuing their hobbies, or interacting socially, while others used emotional detachment and continued working. There is a generalized lack of specific preparation and training for this practice, and sometimes notifiers have taken advantage of their past experiences. Among the forms of support received, the presence of other colleagues during and after the notification of death was usually able to ease the burden of communication.

Most participants communicated the news in person, thus managing to provide practical and emotional support, containing the reactions of the notified, and counteracting their sense of abandonment and loneliness that frequently arises [9,32,33,34]. For some professionals, the exclusive use of the telephone meant avoiding the difficult management of the reactions of family members; for others, the telephone was an obstacle that prevented an expression of closeness. In fact, this tool makes the notification of death impersonal [35] and does not allow for modulating the communication according to the needs of the recipient [36].

The professional figures investigated here used different communication methods: (1) giving detailed communication and explaining the sequence of events; (2) evading the details; (3) using a dry and fast mode; (4) communicating ‘gradually’; (5) extemporaneously, letting the relatives guess what happened. The modalities that emerged reflect the classification of Shaw and colleagues [37], who argued that the three communication styles prevalent in notifiers are: gradual, straightforward and avoidant. The first style consists of detailed and gradual communications; in the second the notifications lacking details are given in a dry and fast manner, and in the third the information is provided ‘extemporaneously’. From the literature we know that using an avoidant style can risk creating greater anxiety, confusion and distress in those who receive the news; on the contrary, a gradual style causes the news to be perceived as more personalized and clear, facilitating a feeling of greater support [37].

Often, professionals perceived a sense of injustice and helplessness during the notification process, probably related to the unexpected and violent nature of the death. This can undermine the sense of security in professionals, causing their world—like the one of the recipients—to collapse too [38]. In addition, many participants felt a sense of inadequacy during the assignment, a feeling that could arise from insufficient training for the task [7]. In fact, adequate training could help professionals feel more prepared and increase their self-esteem [39]. Some professional figures have expressed concern about the long-term effects that the role of death notifier could have; in fact, a continuous and prolonged exposure to the theme of death and tasks of this type could lead to a considerable degree of professional burnout [7] and emotional exhaustion [22,23].

This research confirms that a professional figure in charge of giving the news may have difficulty in aligning with the culture of the notified; this could result in an inability to provide the most personalized support possible. For this reason, it is important that the professional is able to establish an effective intercultural relationship, in order to reduce stereotypes and preconceptions [40,41].

The characteristics of the notified person, such as advanced age, state of health or psychological vulnerability, can also add to the difficulties of a death notification process. The perception of difficulty could arise from the awareness that with these types of recipient, if the communication of death is not carried out properly, it could further aggravate these conditions [32]. For this reason, it could be useful to set up a support group made up of family members and other professionals that may intervene to support the most vulnerable people. The presence of a support group could also be a valuable resource in managing the emotional reactions of the notified persons, which can be particularly critical or severe in the experience of the professionals [12]. In addition, several participants in this study reported that they had difficulty managing their emotional reactions. Two opposing perspectives were observed: for some participants it was important to maintain a certain degree of detachment, in order to offer adequate clarity to the notified; while other participants felt that an empathetic approach was the only way to grasp the real needs of their interlocutor. The presence of these two perspectives is confirmed by the literature, which describes the two positions with the terms rejection and hyper-identification, respectively [34]. The first perspective may be due to the difficulty of dealing with one or more aspects related to the death or the interaction with the notified; therefore, if notifiers cannot delegate the task, they may use a standard, quick and indifferent approach. The hyper-identification response, in turn, may be due to the fact that the notifier has had a similar experience or that there are aspects in common with the notified person (age, cultural background, social or professional position). However, from the literature it emerged that polarized psychological responses can affect the notification process: expressing excessive emotionality or appearing too detached could result in harmful communications [42,43,44]. Therefore, it is important that notifiers are aware of the attitudes they can adopt during the notification process [6]. Furthermore, to cope with the stress of communication, some professionals have implemented different adaptation strategies, including distraction (hobbies, sports, time for oneself). This finding is also supported by the study by Stewart and colleagues [11], who consider distraction as a second-level strategy used in addition to interpersonal support strategies allowing decompressing from the job. Other professionals have used avoidance strategies by not implementing any specific measures to deal with the weight of communication, other than their emotional detachment or the simple continuation of habitual work practice. In this regard, it is possible that professional figures do not have adequate tools to cope with the intensity of emotions or that there are prejudices associated with seeking help in the workplace, causing feelings of isolation and stigmatization [45].

In this survey, most participants—both police officers and health workers—said they had not received any type of training on the subject. Two different positions have been observed in this regard: on the one hand, some professional figures have not received any type of training but consider it necessary; on the other hand, there are professionals that do not even consider it necessary, since the scenarios encountered during the death notification process would be too different from each other. Therefore, professionals in performing this task are guided by common sense, as already found in the literature on communication at the end of life [46]. There was also a share of participants who claimed they had familiarized themselves with the subject of notification of violent and unexpected death thanks to previous experiences, including the vicarious observation of the activities of more experienced colleagues, or having been themselves the recipient of a notification. In the latter case, the risk is that the professional figure may have a hyper-identification response, causing a particularly intense experience in the notified [34]. Receiving support during/after the task from other professional figures such as colleagues, doctors, and nurses seems to be a valid intervention. Apparently, it was useful for the professionals to proactively seek informal support from colleagues, friends, relatives or partners. Sharing similar experiences and reflecting on what has been lived together with a colleague can provide positive validation of the experience, leading to greater self-confidence [47].

The aspects identified in this study seem to have links and connections among them. A possible visualization of these links is presented in Figure 1. How the communication took place, specifically the medium through which it was given, influenced the experience during the communication. For example, for some participants giving the communication by telephone was easier because they did not have close contact with the reactions of those notified; for others, giving the notification at a distance was difficult because they could not provide support. The experience during communication is related to the themes “forms of support” and “coping strategies”: the experience during death communication for many participants was intense, which forced professionals to seek forms of support and implement coping strategies to deal with the burden of the task. The theme “form of support” and specifically the sub-theme “training” are related to the sub-theme “emotion” of the theme “experience during the communication”. In fact, the absence of adequate training means that the professional figure experiences the assignment with a sense of inadequacy. In addition, the sub-theme “training” from past experience is related to the choice of coping strategy adopted, which is very often that of avoidance. Finally, the theme “difficulties encountered” is related to the theme “how the communication took place”; for example, professional figures experienced a sense of helplessness and uselessness when the reactions of those notified were particularly intense.

## 5. Limitations

This study suffers from a number of limitations. Using an ad hoc questionnaire as a data collection tool accessible via an Internet link made was a cost-effective method for reaching the respondents located throughout Italy; however, the questionnaire method inevitably narrowed the responses. The sample of participants cannot be considered as representative of all the experiences of police officers and health workers. Furthermore, in the group of participants, we found an unequal gender distribution, which corresponds to that in the population [30,31], with a prevalence of men among police officers and women among health workers. No comparison analysis was carried out between the group of police officers and that of health workers.

## 6. Conclusions

This study presents the experience of unexpected and violent death notification experienced by Italian police officers and health care workers. The data collected show that for the notifiers, it is a challenging experience due to the strong emotions they experience, as they feel a strong sense of inadequacy and often struggle to manage their emotional load. To deal with this, they implement coping strategies, such as distraction with sports, hobbies and social interactions or sometimes, when involvement is very intense, avoidance strategies. Further complicating the notification process is the adaptation to the characteristics of their interlocutor and the attempt to modulate their linguistic register.

The absence of adequate training on the topic forces professional figures to make use of knowledge from their own past experiences and adopt heterogeneous communication styles. Therefore, in performing this task, professionals are guided by common sense. The elaboration of what was experienced during the assignment is also left to individuals, who sought and appreciated informal discussion, whereas formal interventions of support and advocacy rarely occurred.

It is, therefore, necessary for the performance of this task that professional figures be adequately prepared with curricular, non-prescriptive training that falls into the cultural and working context of the different professions and that formal support and supportive interventions are provided for professionals both during the assignment and once it is over. In addition, it would be useful to establish a multidisciplinary team and improve coordination among the different professional figures who come into contact with the notified person.

The complexities of this sensitive area reinforce the view that more research is needed: it would be desirable to conduct personal interviews to obtain richer and more articulate narratives with a more gender-balanced sample of participants. Future research could focus on comparing the experiences of other professionals who have found themselves reporting unexpected and violent deaths at least once, such as psychiatrists and psychologists.

## Figures and Tables

**Figure 1 ijerph-19-11030-f001:**
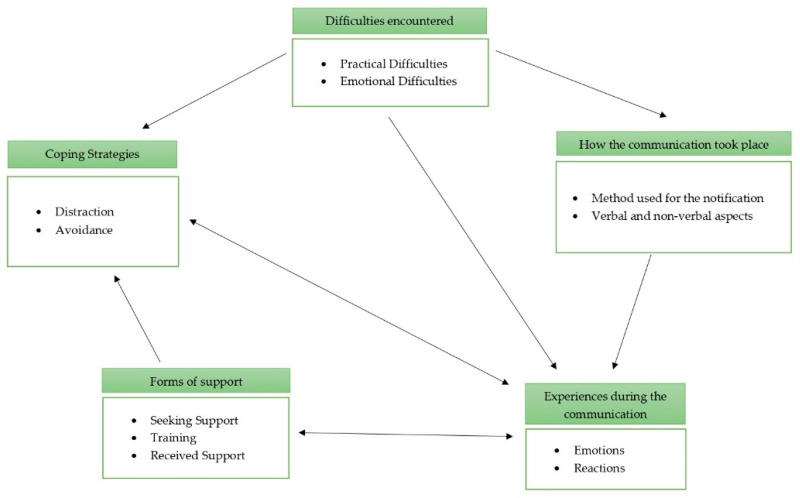
Links among themes.

**Table 1 ijerph-19-11030-t001:** The ad hoc questionnaire.

*Questions to collect demographic information* 1. Age 2. Gender 3. Nationality 4. Family situation 5. Children 6. Profession 7. Year of commencement of professional practice 8. The facility where he/she works is... 9. Department or Service in which it operates 10. Employment status 11. Have you ever reported an unexpected and violent death (suicide, murder, road accident, accident at work, natural disaster)? If so, please specify the number of times………. *Questions about your experience of communicating an unexpected and violent death* 12. How did you communicate the news of an unexpected and violent death? (In person, on the phone, other means…) 13. What did you happen to feel during the communication? 14. What were the main difficulties you encountered? 15. What was most useful to you in dealing with this communication? 16. What measures have you taken to facilitate the receipt of the news? 17. What measures did you take to find relief the burden of communication? 18. Have you received any specific training related to the communication of death? If yes, what kind? 19. Have you received support from others for this type of communication? If yes, what kind? 20. With reference to the COVID-19 pandemic we are experiencing and the restrictions it has entailed, have you ever reported death? If so, what particular experiences did you have in this communication? 21. Would you like to add something? If so, please share your thoughts with us. Thank you!

**Table 2 ijerph-19-11030-t002:** Age, gender and department/service to which the sample belongs.

		*N*	Percentage
*Age*	23–30	12	7.7%
	31–40	21	13.5%
	41–50	54	34.8%
	51–60	62	40.0%
	61–70	5	3.2%
	73	1	0.6%
*Sex*	Females	44	28.4%
	Males	111	71.6%
*Department/service to which the sample belongs*	State Police	98	63.2%
	Local Police	1	0.6%
	Carabinieri	2	1.3%
	118 (Emergency)	23	14.8%
	Emergency Department	9	5.8%
	General Medicine	9	5.8%
	Intensive Care Unit	8	5.2%
	Orthopedics	2	1.3%
	Psychiatry	2	1.3%
	Not stated	1	0.6%

**Table 3 ijerph-19-11030-t003:** Themes, sub-themes, codes and observed frequencies.

Themes	Sub-Themes	Codes	Frequencies
**How the communication took place**			
	Method used		
		In person	117
		Both on the phone and in person	21
		On the phone	9
	Verbal and non-verbal aspects		
		By explaining the sequence of events	26
		Extemporaneous communication	25
		Gradually	22
		Physical and visual contact	21
		Tone of voice	15
		Clarity and simplicity	14
		Details circumvention	13
		Call the deceased by name	13
		Quick communication	9
		Reformulation	9
		Self-disclosure	9
		Organ donation request	7
		Use of past tense verbs	5
		Dressed in uniform	5
		Remembrance of the deceased	4
		Instant nature of death	2
	Where the communication took place		
		Reserved place	12
		Dwelling	7
		Public place	3
	Peculiarities of notification during COVID-19		
		No peculiarities	7
		Cumulative effect	5
		Method	3
		Social distancing	5
		Last farewell impossible	3
**Experiences during the communication**			
	Emotions	Sorry for the notified	75
		Sense of inadequacy	68
		Sense of injustice	24
		Feelings of fear	14
		Fear of long-term effects	12
		Sense of duty	8
		Relief	2
	Reactions		
		Somatic reactions	15
		Crying	10
		Shock	7
**Difficultes encountered**			
	Practical difficulties		
		Cultural and linguistic differences	58
		Notification initiation	15
		Demographic and family characteristics of the notified person	13
		Circumstances of death	12
		Characteristics of the deceased	5
		Characteristics of the notifier	3
		Answer questions without having information	3
		Recognise family members in the crowd	2
		Prohibition to see the body	2
	Emotional difficulties		
		Managing own emotions	75
		Reactions of the notified	33
		No difficulty	14
**Coping strategies**			
	Distraction		
		Sport	14
		Time to reflect	9
		Hobby	8
		Request for days off	1
	Offer of help		
		Further practical support for notified	21
		Further emotional support to the notified	18
	Faith		
		Attribution of meaning to death	7
		Natural course of life	5
	Avoidance		
		No precautions	49
		Emotional detachment	13
		Work practice continuation	3
**Forms of support**			
	Seeking for support		
		Seeking informal support	17
		Seeking formal support	4
	Training		
		No training	107
		Information from previous experiences	39
		Specific training	9
	Received support		
		Support from other people	66
		Debriefing	10
		Sessions with a psychologist	4

## Data Availability

The data presented in this study are available upon reasonable request to the corresponding author. The data are not publicly available, due to their confidential nature.

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
