# Peer review of "Communicating Unexpected and Violent Death: The Experiences of Police Officers and Health Care Professionals"

_ijerph, 2022, doi:10.3390/ijerph191711030_

Round 1
Reviewer 1 Report
Dear authors,
thank you for your manuscript on a very important topic that needs more attention!
I have some comments and suggestions that should be Adresse before your manuscript can be accepted for publication.
1) Introduction:
The introduction is too long and could be shorter. You should present the background and the reason for your study more clear in order to raise the interest of the reader.
State clearly why this study is needed. As the study is part of a wider investigation you could inform the reader short what this wider investigation is about. a source and link would be interesting.
2) Methods:
Where did the study took place? In Italy? This is not clears you have affiliations in Australia, Italy and Slovenia. Please describe more clearly.
It is also needed to write more about your reasons to choose an ad hoc online questionnaire. Why is this the most appropriate method?
I miss a description of the analysis process other than your table from Braun & Clarke. Describe what you have done. How dis you reach consensus about the themes? Who performed the analysis?
3) Discussion
I would appreciate a short summary of the main findings as start of the discussion.
Gender seems to be an important issue in your findings. This should be discussed.
As you have introduced a figure on the links among the themes this should be discussed in more detail.
4) Conclusion
Your conclusion looks more like a discussion. Please highlight the conclusion from your study and your findings. Which implications does it have for future practice and further research? Probably you should move the last part of your limitations. Make the conclusion more clear and define future aims...
Good luck with the revision of your manuscript
Reviewer 2 Report
Anyone can anticipate the difficulties of experts in notifying death, but the explanation of how they suffer is insufficient and superficial. IJERPH's submission rules were not followed, and in particular, the reference notation method was not adapted to this paper. This can be misunderstood as rudeness towards the editor of the paper. Also, the discussion and conclusion part lacks connection with the results and is lengthy. The conclusion did not provide a clear vision. Therefore, it is judged that it is difficult to publish it because it is considered that it is not suitable for this journal.
Round 2
Reviewer 1 Report
Dear authors,
thank you for the revision of the manuscript and the responses to my comments and suggestions.
I think the manuscript has improved!
I have still one important comment:
The conclusion looks like a discussion including references.
Please shorten the conclusion and remove references from the conclusion.
The discussion and the references should be included in the discussion section if needed.
Reviewer 2 Report
Dear authors
Thank you so much for making the revision.
I talked about the reference format, but the part to write the reference list in numerical order at the end has been modified, but the method of writing the author and year in the text is not IJERPH's reference method. Please correct it in such a way that a number is used after the sentence as in [1].
The rest of the content is judged to have followed the recommendations of other reviewers well. There are a lot of shortcomings in the format, but I think it's an appropriate article to present hard-to-find information.
